# Real-time in vivo imaging of extracellular ATP in the brain with a hybrid-type fluorescent sensor

**Nami Kitajima[1], Kenji Takikawa[1], Hiroshi Sekiya[2], Kaname Satoh[1], Daisuke Asanuma[1], Hirokazu Sakamoto[1], Shodai Takahashi[3], Kenjiro Hanaoka[3], Yasuteru Urano[3,4], Shigeyuki Namiki[1], Masamitsu Iino[5], Kenzo Hirose[1,6]\***

[1]Department of Pharmacology, Graduate School of Medicine, The University of Tokyo, Tokyo, Japan; [2]Department of Physiology, Graduate School of Medicine, The University of Tokyo, Tokyo, Japan; [3]Graduate School of Pharmaceutical Sciences, The University of Tokyo, Tokyo, Japan; [4]Graduate School of Medicine, The University of Tokyo, Tokyo, Japan; [5]Department of Cellular and Molecular Pharmacology, Nihon University School of Medicine, Tokyo, Japan; [6]International Research Center for Neurointelligence, The University of Tokyo Institutes for Advanced Study, The University of Tokyo, Tokyo, Japan

**Abstract** Adenosine 5' triphosphate (ATP) is a ubiquitous extracellular signaling messenger. Here, we describe a method for in-vivo imaging of extracellular ATP with high spatiotemporal resolution. We prepared a comprehensive set of cysteine-substitution mutants of ATP-binding protein, *Bacillus* FoF$_1$-ATP synthase $\varepsilon$ subunit, labeled with small-molecule fluorophores at the introduced cysteine residue. Screening revealed that the Cy3-labeled glutamine-105 mutant (Q105C-Cy3; designated ATPOS) shows a large fluorescence change in the presence of ATP, with submicromolar affinity, pH-independence, and high selectivity for ATP over ATP metabolites and other nucleotides. To enable in-vivo validation, we introduced BoNT/C-Hc for binding to neuronal plasma membrane and Alexa Fluor 488 for ratiometric measurement. The resulting ATPOS complex binds to neurons in cerebral cortex of living mice, and clearly visualized a concentrically propagating wave of extracellular ATP release in response to electrical stimulation. ATPOS should be useful to probe the extracellular ATP dynamics of diverse biological processes in vivo.

**\*For correspondence:**
kenzoh@m.u-tokyo.ac.jp

**Competing interests:** The authors declare that no competing interests exist.

## Introduction

Adenosine 5' triphosphate (ATP) is the essential energy currency of intracellular metabolism of all living systems, but also plays important roles as an extracellular signaling molecule in a wide variety of physiological and pathological processes, including neurotransmission (*Burnstock, 2007*), neuron–glia signaling (*Fields and Burnstock, 2006*; *Khakh and North, 2012*), tissue circulation (*Cheng et al., 2018*; *González-Alonso et al., 2002*; *MacVicar and Newman, 2015*), regulation of the immune system (*Idzko et al., 2014*; *Junger, 2011*), and cancer development (*Di Virgilio et al., 2018*; *Stagg and Smyth, 2010*). In order to elucidate the mechanisms of these processes, an understanding of the spatiotemporal dynamics of ATP is essential. To date, a variety of techniques for measurement of ATP have been developed: bioluminescence assay methods using firefly luciferase (*Holton, 1959*; *Pellegatti et al., 2005*; *Wang et al., 2000*), electrochemical methods including voltammetry (*Singhal and Kuhr, 1997*), enzyme-coupled electrode techniques (*Kueng et al., 2004*; *Llaudet et al., 2005*), cell-based electrophysiological techniques (*Brown and Dale, 2002*; *Pangrsic et al., 2007*), and whole cell-based fluorescence assay methods using Ca$^{2+}$ indicator (*Anselmi et al., 2008*; *Huang et al., 2007*). Some of them have been applied to measure

**eLife digest** Biologists often refer to a small molecule called adenosine triphosphate – or ATP for short – as 'the currency of life'. This molecule carries energy all through the body, and most cells and proteins require ATP to perform their various roles.

Nerve cells (also known as neurons) in the brain release ATP when activated, and use this molecule to send signals to other active neurons or other cells in the brain. But ATP can also signal danger in the brain. A molecule derived from ATP is involved in transmitting the pain signals of migraines and severe headaches; and ATP levels can become imbalanced after strokes, when parts of the brain are deprived of blood.

Despite its importance, ATP remains difficult to visualize in the body, and monitoring the molecule in the active brain in real time is challenging. To address this issue, Kitajima et al. designed an optical sensor that could monitor ATP in the healthy brain, and was sensitive enough to detect when and where it was released. First, Kitajima et al. made several potential sensors by attaching various fluorescent tags to different locations on a protein that binds ATP. Next each sensor was tested to determine whether it could bind ATP tightly and get bright upon binding. This is important because previous sensors could not detect ATP release in the brains of living animals.

To illustrate the new sensors' potential, Kitajima et al. used the sensor to image ATP in the brains of live mice. A 'wave' of ATP was seen spreading through the brain after neurons were stimulated with a small electric pulse, mimicking a sudden migraine or stroke.

The results confirm that this new sensor is suitable for imaging how ATP signals in the brain, and it may help resolve the underlying mechanisms of migraines and strokes. This sensor could also be used to understand other cellular process which rely on ATP to carry out their role.

extracellular ATP concentrations in vivo (*Gourine et al., 2005*; *Melani et al., 2005*; *Pellegatti et al., 2008*). For example, bioluminescence assay is able to quantify the extracellular ATP levels in micro-dialysis samples (*Melani et al., 2005*), and enzyme-coated electrodes inserted in tissues allow continuous monitoring of extracellular ATP levels in the brain (*Gourine et al., 2005*). However, these methods do not provide spatial information. Imaging of the bioluminescence generated by luciferin-luciferase reaction might be a solution for spatially resolved measurement of ATP (*Rajendran et al., 2016*), and indeed, engineered cells displaying luciferase on the plasma membrane were used to visualize extracellular ATP in vivo by inoculating the cells into living mice (*Pellegatti et al., 2008*). However, both spatial resolution and temporal resolution in luciferase-based imaging are inherently compromised by the low photon flux of bioluminescence emission (*Rajendran et al., 2016*).

On the other hand, imaging techniques using fluorescent sensors can provide excellent spatio-temporal resolution (*Giepmans et al., 2006*), and a variety of fluorescent ATP sensors based on fluorescent proteins have been developed (*Arai et al., 2018*; *Berg et al., 2009*; *Imamura et al., 2009*; *Lobas et al., 2019*; *Tantama et al., 2013*; *Yaginuma et al., 2015*). These genetically encoded ATP sensors can visualize intracellular ATP dynamics, and have facilitated our understanding of ATP signaling inside cells (*Depaoli et al., 2018*; *Zala et al., 2013*). In contrast, attempts to adapt the existing sensors for extracellular use have achieved limited success, and in vivo imaging of extracellular ATP is still difficult (*Conley et al., 2017*; *Lobas et al., 2019*). One problem is that the affinities of these extracellular ATP sensors are too low to monitor extracellular ATP, the concentration of which likely ranges from hundreds of nanomolar to micromolar levels (*Yegutkin, 2008*). Moreover, these extracellular ATP sensors are designed to be synthesized inside the cells and transported to the plasma membrane to face the extracellular space (*Conley et al., 2017*; *Lobas et al., 2019*). Imperfect membrane transport as well as potential posttranslational modifications such as glycosylation during this process can compromise the performance of the sensors (*Morciano et al., 2017*). Another problem is the strong pH dependency of these genetically encoded ATP sensors, due to the innate pH sensitivity of the fluorescent proteins used (*Chudakov et al., 2010*; *Tsien, 1998*). This pH dependence is especially problematic in in vivo ATP imaging, because changes in ATP concentration in tissues are often accompanied by pH fluctuations (*Berg et al., 2009*; *Di Virgilio et al., 2018*; *Gourine et al., 2010*).

Another strategy to develop fluorescent sensors is a hybrid-type probe design in which a small-molecule fluorescent dye is conjugated to a ligand-binding protein (*Takikawa et al., 2014*; *Wang et al., 2009*). We previously developed a hybrid-type glutamate sensor, in which the AMPA receptor GluR2 subunit is employed as a glutamate-binding protein (*Namiki et al., 2007*), and visualized extrasynaptic glutamate dynamics in vivo (*Okubo et al., 2010*). In this study, to develop a hybrid-type ATP sensor, we adopted bacterial FoF$_1$-ATP synthase ε subunit as an ATP-binding protein, because fluorescent-dye conjugation has been used to study conformational changes of the protein upon ATP binding (*Iino et al., 2005*; *Kato et al., 2007*). We screened fluorescent dyes and their labeling sites within the ATP-binding protein, and obtained a highly sensitive ATP sensor with a large fluorescence response, high selectivity for ATP, and insensitivity to pH fluctuation, suitable for detection of extracellular ATP. The sensor can be anchored on the plasma membrane to detect extracellular ATP. To illustrate potential applications of the sensor, we performed extracellular ATP imaging in the brain of living mice, and observed the propagation of a wave-like extracellular ATP increase through the cerebral cortex in response to neuronal excitation evoked by electrical stimulation. The results confirm that the ATP sensor is suitable for in vivo imaging of the dynamics of extracellular ATP signaling.

## Results

### Generation of high-sensitivity ATP sensors for extracellular ATP imaging

To visualize extracellular ATP, we designed a hybrid-type fluorescent ATP sensor composed of an ATP-binding protein and a small-molecule fluorescent dye. We adopted the ε subunit of FoF$_1$-ATP synthase of thermophilic *Bacillus* PS3, which has high affinity for ATP (*Kato-Yamada and Yoshida, 2003*), as the ATP-binding protein, and introduced a single cysteine substitution in its amino-acid sequence to enable the conjugation of a small-molecule fluorescent dye bearing a cysteine-reactive maleimide group (*Figure 1A*). The performance of a hybrid-type sensor is determined by the location of the fluorophore within the ligand-binding protein (*Namiki et al., 2007*; *Takikawa et al., 2014*). To obtain a high-sensitivity ATP sensor, we used our previously developed high-throughput screening system, HyFInD (*Takikawa et al., 2014*), in which a large variety of fluorescent conjugates can be generated by varying both the fluorophore conjugation sites and the fluorophores, and then evaluated for fluorescence response to a ligand. We generated 134 cysteine mutants covering every position in the amino-acid sequence except for the first position (the initiating methionine). We used four small-molecule fluorescent dyes, Oregon Green (OG), Alexa Fluor 488 (Alexa488), Cy3 and SiR650, resulting in the generation of 536 fluorescent conjugates in total. For the initial screening we employed a high concentration (2 mM) of ATP to select sensors having a large dynamic range of fluorescence response. For each of the four fluorophores, we found a moderate number of fluorescent conjugates showing detectable fluorescence changes upon application of ATP, but with a large variation in the response (*Figure 1B*). Notably, even if the fluorophore labeling site was the same, the fluorescence response varied depending on the type of fluorophore.

We then selected eight fluorescent conjugates as candidate sensors showing a large fluorescence response in the initial screening, and tested their fluorescence response to a low ATP concentration (200 nM) (*Figure 1—figure supplement 1A*). Variation in the fluorescence response was again observed, as in the case of the higher concentration of ATP. The conjugate exhibiting the largest fluorescence response at 200 nM ATP was the Cy3-labeled cysteine substitution mutant at glutamine 105 (Q105C-Cy3). We hereafter designate Q105C-Cy3 as ATPOS (<u>A</u>TP <u>O</u>ptical <u>S</u>ensor).

We next examined the characteristics of ATPOS in more detail. In the absence of ATP, ATPOS displayed a fluorescence excitation peak at 556 nm and an emission peak at 566 nm (*Figure 1C*; *Table 1*). At 1 mM ATP, the fluorescence intensity of ATPOS increased without any appreciable change in the peak wavelengths (excitation peak at 556 nm, emission peak at 564 nm), while the absorption spectrum showed almost no change upon the addition of ATP (*Figure 1—figure supplement 1B*). On the other hand, the fluorescence quantum yield increased from 0.29 to 0.62 (*Table 1*). The results indicate that the change in fluorescence intensity is due to the change in the quantum yield. To estimate the sensitivity of ATPOS for ATP, we examined the dependence of the fluorescence change of ATPOS on the concentration of ATP (*Figure 1D*). Fluorescence changes were

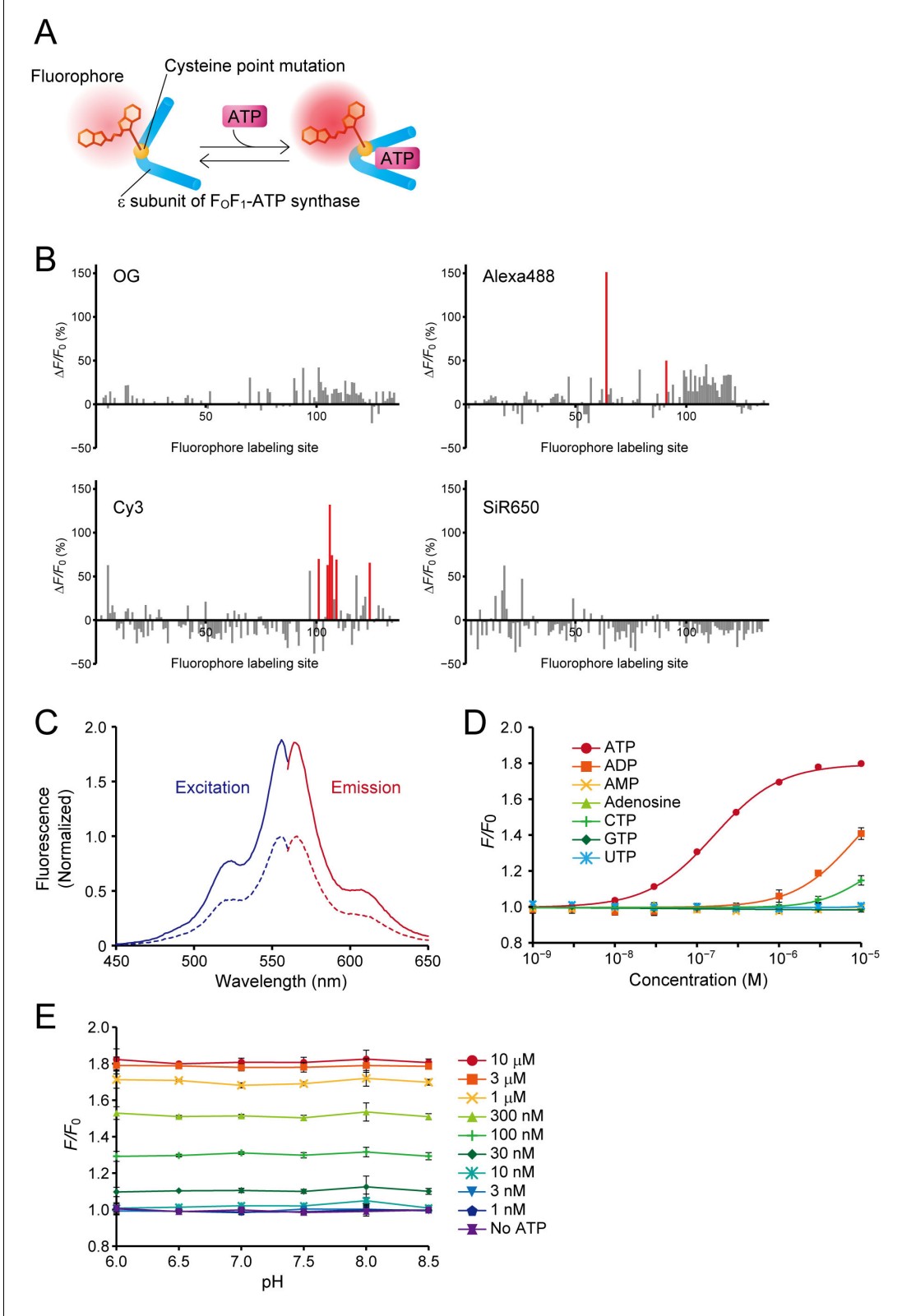

**Figure 1.** Design, optimization, and characterization of ATPOS in vitro. . (**A**) Schematic illustration of a hybrid-type ATP sensor composed of the ε subunit of FoF1-ATP synthase having a cysteine point mutation at which a small-molecule fluorophore is conjugated. (**B**) Results of HyFInD screening. Bars indicate fluorescence changes ($\Delta F/F_0$) of fluorescence conjugates labeled with OG (upper left), Alexa488 (upper right), Cy3 (bottom left), and SiR650 (bottom right) in response to 2 mM ATP. The numbers 1–134 on the horizontal axis represent the positions in the amino-acid sequence at which

*Figure 1 continued on next page*

*Figure 1 continued*

cysteine point mutation is introduced for fluorophore labeling. The fluorescent conjugates indicated by red bars were tested in the second screening. (C) Normalized excitation (blue) and emission spectra (red) of ATPOS with (solid line) or without 1 mM ATP (dashed line). (D) Dose-response curves of ATPOS to ATP, ADP, AMP, adenosine, CTP, GTP, and UTP (n = 3). Plots were fitted with the Hill equation. (E) The pH dependence of ATPOS. Fluorescence intensities normalized to the fluorescence intensity at pH 7.4 without ATP ($F/F_0$) are plotted as a function of pH (n = 3). Data are presented as mean ± SEM. In some figures, error bars representing SEM are smaller than the symbols indicating the mean.

The online version of this article includes the following figure supplement(s) for figure 1:

**Figure supplement 1.** Screening and characterization of ATPOS.

observed at submicromolar concentrations, and fitting the Hill equation to the dose-response data yielded an apparent dissociation constant of ~150 nM and a Hill coefficient of 1.08. The results suggest that ATPOS has sufficiently high affinity for ATP to be applicable for the visualization of extracellular ATP. The Hill coefficient value of ~1 suggests that ATPOS binds to ATP with 1:1 stoichiometry. We also compared the affinity for ATP with those for ATP metabolites and other nucleotides to examine the selectivity of ATPOS. The affinity for ATP was about one hundred-fold higher than that for ADP, and ATPOS was almost insensitive to other metabolites of ATP, such as AMP and adenosine, and other nucleotides such as GTP, CTP and UTP (*Figure 1D*). To study the kinetics of ATPOS, we performed stopped-flow measurements (*Figure 1—figure supplement 1C,D and E*). The observed rate constants showed a dose-dependent increase ranging from 5.0 $s^{-1}$ at 0.5 µM ATP to 8.9 $s^{-1}$ at 2.0 µM ATP. We also measured fluorescence decay of ATPOS upon a decrease in ATP concentration, and determined the decay rate constant to be 2.0 $s^{-1}$. These results suggest that the association and dissociation kinetics of ATPOS is fast enough to monitor ATP dynamics on a time scale of seconds.

The pH dependence of ATP sensors is a critical determinant of the accuracy of ATP quantification, because pH fluctuations often occur at the same time as changes in ATP concentration (*Berg et al., 2009*; *Di Virgilio et al., 2018*; *Gourine et al., 2010*). Therefore, we assessed the pH sensitivity of ATPOS and observed no fluorescence response to changes in pH from 6.0 to 8.5 in the presence or absence of ATP (*Figure 1E*).

## Introduction of anchoring ability on the surface of cultured neurons and capability for ratiometric measurements

For in vivo ATP imaging with ATPOS, we introduced two additional features. Firstly, in order to anchor ATPOS to the outer surface of cell membranes, enabling it to report extracellular ATP in brain tissue, we employed BoNT/C-Hc, a nontoxic subunit of *Clostridium botulinum* neurotoxin, which binds to the plasma membrane of neurons (*Tsukamoto et al., 2005*). Secondly, we introduced a capability for dual-color ratiometric imaging, which should mitigate the effect of spatial inhomogeneity of sensor concentration, as well as movement artifacts. For this purpose, we employed Alexa488 as a fluorophore for a reference color channel. Accordingly, we assembled a molecular complex of ATPOS with BoNT/C-Hc and Alexa488 labeled-streptavidin (*Figure 2A*). The ATPOS complex exhibited a large fluorescence change in the red (Cy3) channel and high sensitivity for ATP, comparable to those of ATPOS itself, whereas it showed no fluorescence change in the green (Alexa488) channel (*Figure 2—figure supplement 1*). Therefore, the ratio of red to green fluorescence provides a measure of ATP concentration.

We next tested the performance of the ATPOS complex anchored to cellular membranes using primary cultures of rat hippocampal neurons. After the ATPOS complex was applied, followed by

**Table 1.** In vitro properties of ATPOS.

|  | $\lambda_{abs,max}$ (nm) | $\lambda_{em,max}$ (nm) | QY |
|---|---|---|---|
| Cy3 | 550 | 570 | 0.04 |
| Cy3B | 558 | 572 | 0.67 |
| ATPOS (0 mM ATP) | 551 | 566 | 0.29 |
| ATPOS (1 mM ATP) | 552 | 564 | 0.62 |

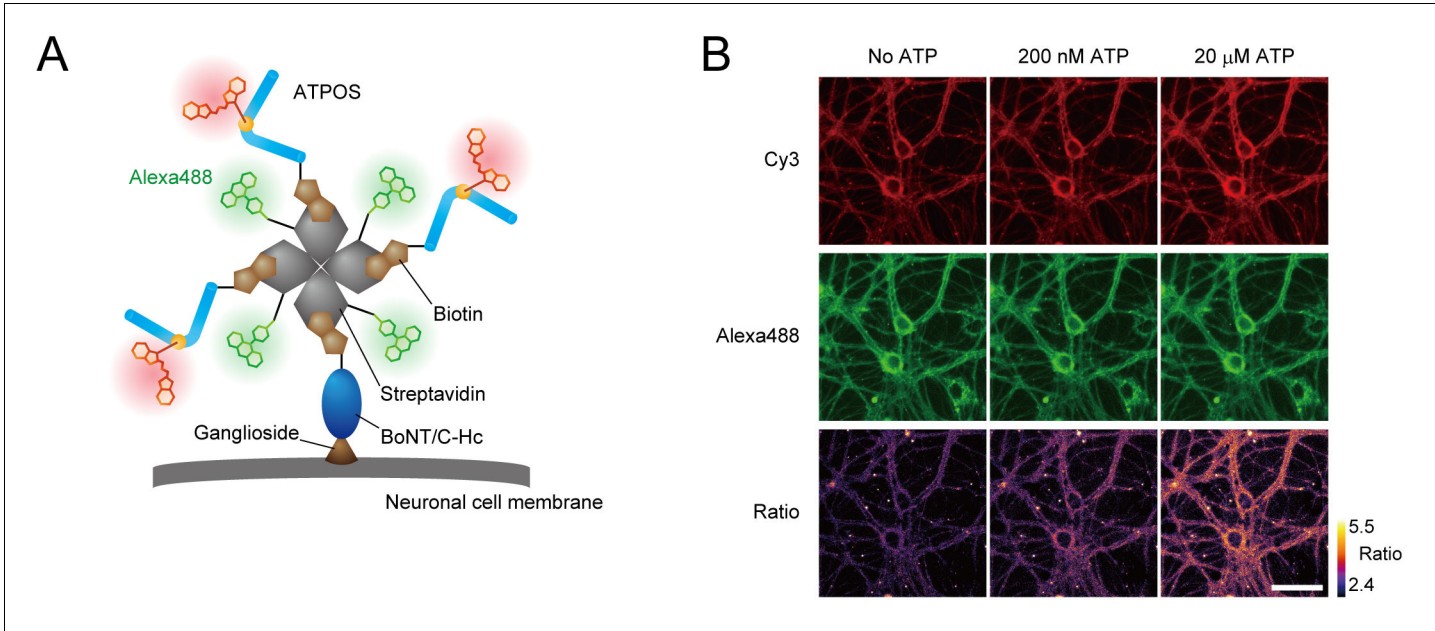

**Figure 2.** Anchoring of ratiometric ATPOS on the surface of cultured neurons. (**A**) Strategy for anchoring of ratiometric ATPOS on neuronal surfaces. ATPOS is tethered to neuronal cells by BoNT/C-Hc, which binds to neuronal surface gangliosides. ATPOS and BoNT/C-Hc are coupled via a biotin-streptavidin linkage. Streptavidin labeled with Alexa488 is used for ratiometry. The scheme illustrates the ideal form of the ATPOS complex, assuming that a mixing molar ratio (3:1:1) of ATPOS, streptavidin and BoNT/C-Hc is preserved. The actual stoichiometry of the components may differ from that in the scheme. (**B**) Representative images of cultured neurons labeled with the ATPOS complex in the absence (left) or presence of 200 nM (middle) or 20 µM ATP (right). Scale bar, 50 µm.

The online version of this article includes the following figure supplement(s) for figure 2:

**Figure supplement 1.** Sensitivity of ATPOS complex for ATP in vitro.

wash-out, both red and green fluorescence was observed along the neurons (*Figure 2B*), indicating successful anchoring of the ATPOS complex on the neuronal cell surface. The fluorescence intensity in the red channel increased upon ATP addition, but that in the green channel did not; consequently the fluorescence ratio was increased. These results indicate that the ATPOS complex anchored on the neuronal surface can work as a ratiometric sensor of extracellular ATP.

## In vivo imaging of extracellular ATP in the cerebral cortex

We then applied the ATPOS complex for extracellular ATP imaging in the cerebral cortex of living mice. The ATPOS complex was pressure-injected into layer 2/3 of the cerebral cortex with a fine glass pipette through a craniotomy (*Figure 3A*). The area labeled by ATPOS, identified by its fluorescence, gradually spread across the cortex to occupy a region 2 mm in diameter (*Figure 3—figure supplement 1A*). The fluorescence of ATPOS was stably observed after the end of the injection. To examine the distribution of ATPOS in the cerebral cortex with cellular resolution, we performed two-photon imaging of the target cortical area, and found that the blood vessels and cell bodies remained unstained (*Figure 3—figure supplement 1B*). These results indicate that ATPOS is immobilized in the extracellular space of the cortex.

To confirm that ATPOS introduced into the cerebral cortex in this way works as a sensor of extracellular ATP, we next injected ATP into the cortex with a glass pipette, and visualized the ratiometric fluorescence response of ATPOS by means of wide-field fluorescence microscopy. Injection of ATP elicited an increase of the fluorescence ratio to a peak amplitude of $6.5 \pm 0.6\%$ (n = 18 from five mice) (*Figure 3B and C*; *Figure 3—videos 1* and *2*). To check the selectivity of ATPOS's response, we applied apyrase, which sequentially hydrolyzes ATP to ADP and ADP to AMP, to the cortex, and observed whether it reduced the fluorescence response of ATPOS to the ATP injection. In the presence of apyrase, the peak amplitude of the ratio was reduced to $2.4 \pm 0.5\%$ (n = 14 from three mice), while a sham injection containing bovine serum albumin had almost no effect (*Figure 3D and*

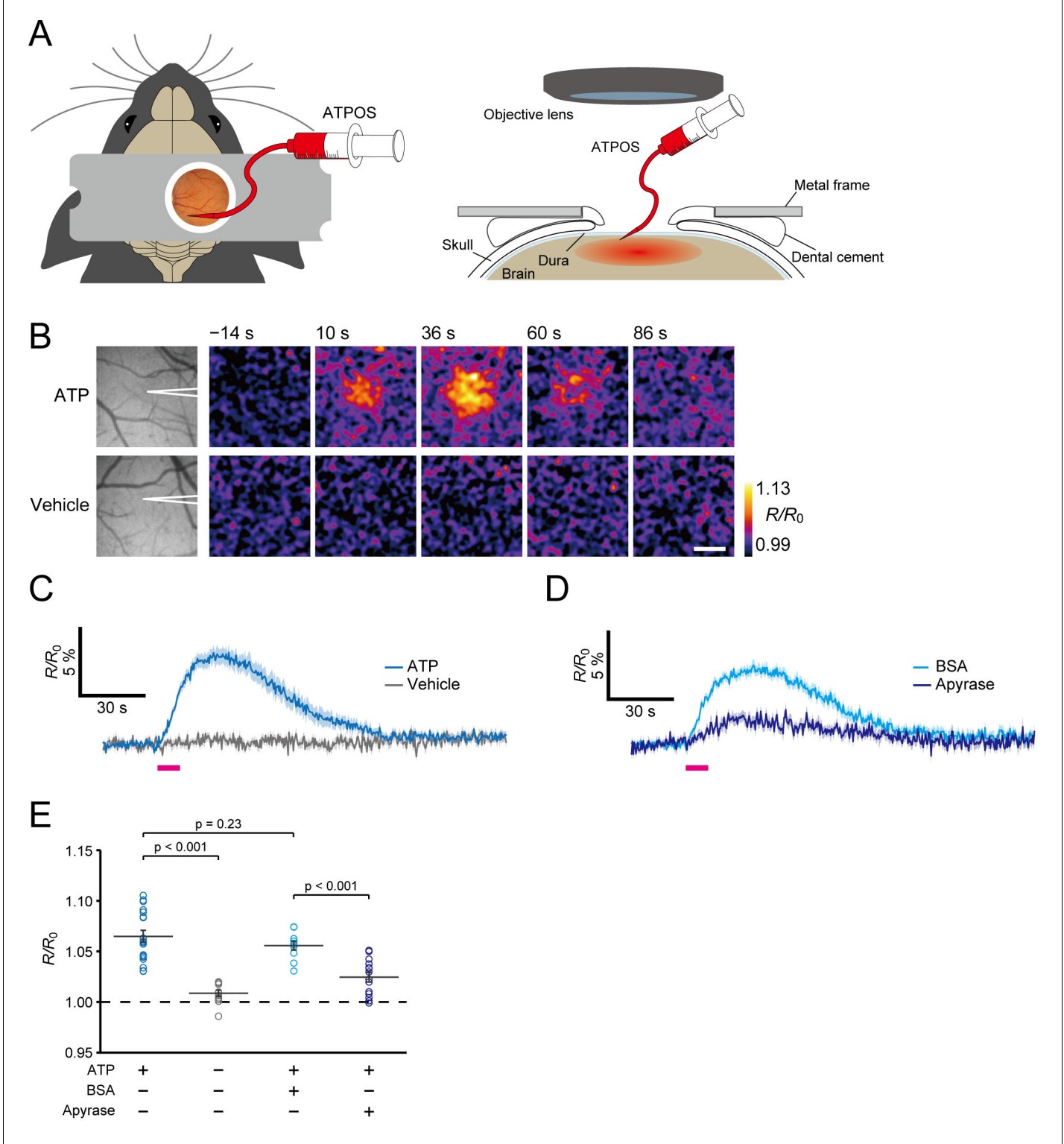

**Figure 3.** Imaging of extracellular ATP in mouse cerebral cortex. (**A**) Schematic illustration of application of ATPOS to the mouse cerebral cortex: top (left) and side (right) views. ATPOS is pressure-injected with a fine glass pipette through a cranial window. (**B**) Representative images showing the ratiometric fluorescence response ($R/R_0$) of ATPOS upon application of 10 mM ATP (top) or vehicle (bottom). The position of the glass pipette for ATP injection is depicted in the left panels. Time after the start of injection is presented above the images. Scale bar, 250 μm. (**C**) Averaged traces of the ratiometric fluorescence response ($R/R_0$) of ATPOS upon application of 10 mM ATP (blue; n = 18 from five mice) or vehicle (gray; n = 11 from three mice). Magenta bar indicates the application of ATP or vehicle. Shaded regions represent SEM. (**D**) Averaged traces of the ratiometric fluorescence

*Figure 3 continued on next page*

*Figure 3 continued*

response ($R/R_0$) of ATPOS upon application of 10 mM ATP (magenta bar) in the presence of bovine serum albumin (BSA) (light blue; n = 10 from three mice) or apyrase (dark blue; n = 14 from three mice). Shaded regions represent SEM. (E) The peak amplitudes of the ratiometric fluorescence response ($R/R_0$) of ATPOS upon application of ATP (blue) or vehicle (gray) in the absence or presence of BSA (light blue) or apyrase (dark blue). Wilcoxon's rank-sum test was performed. Data are presented as mean ± SEM.

The online version of this article includes the following video and figure supplement(s) for figure 3:

**Figure supplement 1.** Application of ATPOS to the mouse cerebral cortex.

**Figure 3—video 1.** Ratiometric fluorescence response of ATPOS upon application of ATP.

https://elifesciences.org/articles/57544#fig3video1

**Figure 3—video 2.** Ratiometric fluorescence response of ATPOS upon application of vehicle.

https://elifesciences.org/articles/57544#fig3video2

---

*E*). These results support the view that ATPOS applied in the mouse cortex does indeed report the extracellular ATP concentration.

## Visualization of extracellular ATP signaling during neuronal excitation

We then attempted to visualize extracellular ATP endogenously released during neuronal excitation in living mice. Previous reports indicate that a propagating wave of neuronal depolarization evoked by electrical stimulation or KCl application to the brain (*Pietrobon and Moskowitz, 2014*) causes elevation of the extracellular ATP concentration (*Heinrich et al., 2012*; *Schock et al., 2007*). Accordingly, we delivered electrical stimulation to the cortex with a metal microelectrode to induce a wave of neuronal depolarization, and observed the fluorescence response of ATPOS during the neuronal excitation under a wide-field microscope (*Figure 4A*). The ratiometric images show a wave-like propagation starting around the electrode (*Figure 4B and C*; *Figure 4—video 1*). A precipitous rise of the fluorescence intensity is seen at the wave front, and the increase shifts in a radial direction away from the stimulation site (*Figure 4D*). The increase of the ratio, which showed a peak amplitude of ~30%, lasted about 2 min. To verify that this wave-like propagation represented the dynamics of extracellular ATP, we compared the fluorescence responses of ATPOS to the stimulation before and after the application of apyrase. In the presence of apyrase, although a wave-like propagation could still be discerned (*Figure 4E and F*; *Figure 4—video 2*), its peak amplitude was substantially reduced to 15.8 ± 2.5% (n = 7 from five mice) while that of the controls was 31.4 ± 1.4% (n = 11 from five mice) (*Figure 4G and H*). These results suggest that the wave-like propagation of fluorescence changes of ATPOS reflects the dynamics of extracellular ATP signaling during neuronal excitation.

In addition, to visualize the dynamics of extracellular ATP during the neuronal excitation at a single cross section, we performed two-photon imaging of ATPOS upon electrical stimulation in layer 2/3, where ATPOS was injected (*Figure 4—figure supplement 1*). The wave-like propagation of fluorescence changes of ATPOS was also observed at a single cross section in layer 2/3. These results are consistent with those obtained by wide-field microscopy, showing the wave-like propagation of extracellular ATP during neuronal excitation.

## Discussion

In the present study, we developed a fluorescent ATP sensor that enables real-time imaging of extracellular ATP dynamics in vivo, and we showed that this sensor, ATPOS, can visualize wave-like extracellular ATP signaling in the cerebral cortex of living mice in response to electrical stimulation.

ATPOS has at least four advantages for in vivo visualization of extracellular ATP. First, ATPOS has a very high affinity for ATP; its apparent dissociation constant is ~150 nM, while those of previously reported fluorescent ATP sensors, such as ATeam, QUEEN and iATPSnFR, are in the range of several micromolar to several millimolar (*Arai et al., 2018*; *Conley et al., 2017*; *Imamura et al., 2009*; *Lobas et al., 2019*; *Yaginuma et al., 2015*). The low affinity of the conventional sensors makes them unsuitable for in vivo extracellular ATP imaging. Second, ATPOS displays high brightness; its fluorescence quantum yield (QY) in the presence of ATP was around 0.62, while that of Cy3 itself, which is employed in ATPOS, is only around 0.04 (*Table 1*). The QY of ATPOS is equivalent to that of Cy3B (QY = 0.67), a rigidified analog of Cy3, which was designed to prevent nonradiative energy loss due

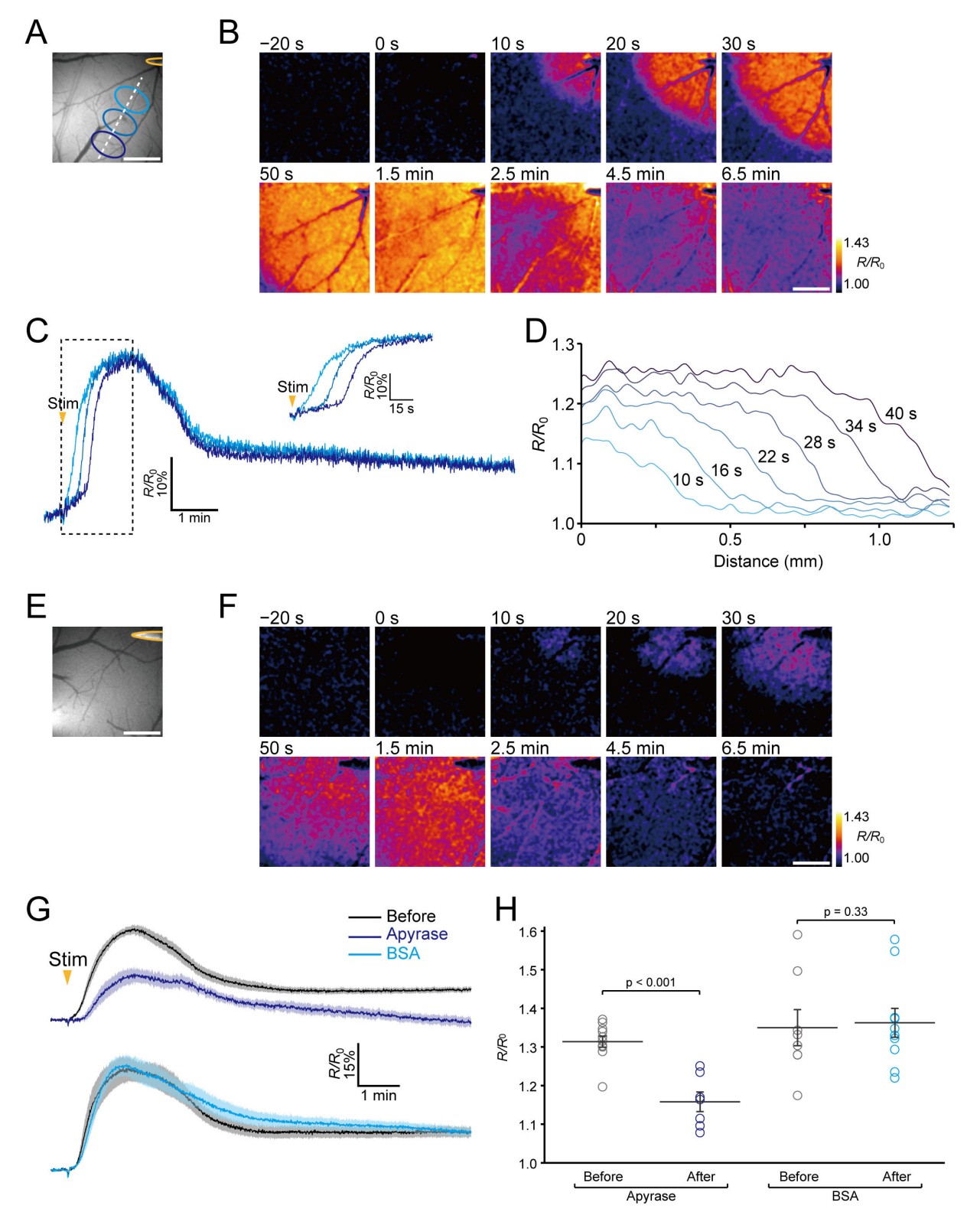

**Figure 4.** Imaging of extracellular ATP signaling during neuronal excitation. (**A**) The position of the stimulating electrode (yellow) and locations of ROIs. (**B**) Representative images of the ratiometric fluorescence response ($R/R_0$) of ATPOS to electrical stimulation. The time after the initiation of stimulation is presented above the images. Scale bar, 500 μm. (**C**) Time courses of the ratiometric fluorescence response ($R/R_0$) of ATPOS extracted from the ROIs depicted in (**A**). The inset shows a magnified view. (**D**) The spatial distribution of ratiometric fluorescence response ($R/R_0$) of ATPOS along the line

*Figure 4 continued on next page*

*Figure 4 continued*

depicted in (**A**). The time relative to the stimulation onset is presented above the lines. (**E**) The position of the stimulating electrode. (**F**) Images of the ratiometric fluorescence response ($R/R_0$) of ATPOS upon electrical stimulation in the presence of apyrase. Time after the initiation of stimulation is presented above the images. Scale bar, 500 μm. (**G**) Averaged traces of the ratiometric fluorescence response ($R/R_0$) of ATPOS to electrical stimulation before (black; n = 11 from five mice) and after the application of apyrase (dark blue; n = 7 from five mice) (upper), and before (black; n = 8 from four mice) and after the application of BSA (light blue; n = 10 from four mice) (lower). Shaded regions represent SEM. (**H**) The peak amplitudes of the ratiometric fluorescence response ($R/R_0$) of ATPOS upon electrical stimulation before (black) or after the application of apyrase (dark blue) or BSA (light blue). Wilcoxon's rank-sum test was performed. Data are presented as mean ± SEM.

The online version of this article includes the following video and figure supplement(s) for figure 4:

**Figure supplement 1.** Two-photon imaging of extracellular ATP signaling during neuronal excitation.

**Figure 4—video 1.** Ratiometric fluorescence response of ATPOS to electrical stimulation.
https://elifesciences.org/articles/57544#fig4video1

**Figure 4—video 2.** Ratiometric fluorescence response of ATPOS to electrical stimulation in the presence of apyrase.
https://elifesciences.org/articles/57544#fig4video2

to rotational and translational vibration modes (*Cooper et al., 2004*). Therefore the high brightness of ATPOS may result from restriction of the intramolecular vibration modes of Cy3 within ATPOS. Third, ATPOS shows high selectivity for ATP over ATP metabolites and other nucleotides. Therefore, compared with conventional fluorescent sensors targeting adenine compounds (*Tokunaga et al., 2012*) or nucleoside polyphosphates (*Kurishita et al., 2012*), ATPOS should be better suited to study the specific roles of extracellular ATP, because derivatives of ATP also work as extracellular signaling molecules (*Dunwiddie and Masino, 2001*). This characteristic of ATPOS is likely attributable to the ligand selectivity of the ε subunit of FoF1-ATP synthase (*Kato-Yamada and Yoshida, 2003*), because existing ATP sensors with high selectivity, including ATeam, QUEEN and iATPSnFR, also adopt the ε subunit as an ATP-binding protein (*Arai et al., 2018*; *Conley et al., 2017*; *Imamura et al., 2009*; *Lobas et al., 2019*; *Yaginuma et al., 2015*). Lastly, ATPOS is insensitive to pH changes. Existing fluorescent protein-based ATP sensors all show pH-dependent changes of fluorescence intensity (*Arai et al., 2018*; *Berg et al., 2009*; *Imamura et al., 2009*; *Lobas et al., 2019*; *Tantama et al., 2013*; *Yaginuma et al., 2015*). The pH insensitivity of ATPOS can be ascribed to its hybrid-type design, in which a pH-insensitive small-molecule fluorescent dye is employed instead of a fluorescent protein. This feature of ATPOS enables precise quantitative analysis of ATP dynamics regardless of pH fluctuations. Although pH in the brain is homeostatically regulated, pH fluctuations are known to occur concomitantly with neuronal activity (*Chesler and Kaila, 1992*; *Magnotta et al., 2012*). Furthermore, the brain pH can change drastically in pathological conditions such as ischemia (*Nedergaard et al., 1991*). Thus, its pH insensitivity greatly extends the applicability of ATPOS to various biological conditions.

Despite these great advantages over genetically encoded sensors, there are drawbacks in the use of ATPOS. One is that the injection procedure required for delivery of ATPOS might cause tissue damage. Although this possibility should be taken into account, we suppose that the tissue damage is likely to be limited, given that the glass micropipette used for the injection procedure is similar in shape to glass electrodes conventionally used in electrophysiological experiments. Another drawback is that targeting ATPOS to specific cell types is not readily feasible. We anticipate that combining technology for cell type-specific display of tag proteins might enable the specific targeting of ATPOS (*Hinner and Johnsson, 2010*).

The screening of fluorophore labeling sites enabled us to select ATP sensors exhibiting a large fluorescence response to ATP. The maximal fluorescence response of ATPOS to ATP is about 2-fold, which is sufficient for the detection of ATP dynamics in vivo. Some ATP sensors found through the screening exhibited a large fluorescence change, but showed low sensitivity. For example, the Alexa488-labeled sensor in which serine-64 was replaced by cysteine (S64C-Alexa488), whose $EC_{50}$ was tens of micromolar, showed about 200% maximal fluorescence response to ATP. These low-affinity ATP sensors obtained in the screening may be useful for ATP imaging under circumstances where the ATP concentration is elevated to such high levels. We also obtained different-colored ATP sensors displaying green or near-infrared fluorescence. This rich color variation should enable multiplexed imaging of ATP together with $Ca^{2+}$ and other signaling molecules, and should also be compatible with the manipulation of cellular activities by means of optogenetics.

ATPOS revealed the dynamic response of extracellular ATP to electrical stimulation (10 mA for 1 s) in the cerebral cortex of living mice. Such electrical stimulation has been used to evoke a propagating wave of neuronal depolarization (*Reid et al., 1987*), known as cortical spreading depression (CSD), which is involved in the pathogenesis of migraine and stroke (*Lauritzen et al., 2011*). We found that the elevation of extracellular ATP around the stimulation site propagates through the cortex in a concentric fashion. Although the elevation of extracellular ATP concentration during CSD is already known (*Heinrich et al., 2012*; *Schock et al., 2007*), its spatial dynamics have not been reported. We found that the ATP wave propagated at the speed of ~2 mm/min, which is comparable to the reported rate of neuronal propagation of CSD, 2 to 5 mm/min (*Khennouf et al., 2016*; *Takagaki et al., 2014*; *van den Maagdenberg et al., 2004*), suggesting a role of the ATP-release mechanism in CSD. The precipitous rise of ATP at the wave front suggests that the ATP wave is likely to be formed by active propagation of ATP release, rather than passive diffusion. In addition, the ATP wave was observed in the entire field of view, suggesting that the elevation of ATP concentration occurs across a broad area of the brain. Given that ATP works as a signaling molecule for both neurons and glial cells (*Burnstock, 2007*; *Fields and Burnstock, 2006*; *Khakh and North, 2012*), the ATP wave during CSD may play an important role in the cellular processes involved in the pathogenesis of migraine and stroke (*Charles and Baca, 2013*; *Pedata et al., 2016*). We anticipate that ATPOS will be a useful tool not only for studies of these pathophysiological processes, but also for investigating the feasibility of novel clinical treatments targeting ATP signaling.

## Materials and methods

### Animal experiments

All procedures used in animal experiments were in accordance with the guidelines established by the Animal Welfare Committee of the University of Tokyo.

### Cell culture

Hippocampal neuronal cultures were prepared from Sprague-Dawley rats (SLC; embryonic, 21 days old). The embryos were decapitated, and the hippocampi were dissected out and treated with trypsin (Life Technologies) and DNase I (Sigma). The dissociated neurons were incubated on glial cell monolayers plated on coverslips coated with laminin (Life Technologies) and poly-L-lysine (Nacalai tesque) in Neurobasal A medium (Life Technologies) supplemented with B-27 supplement (Life Technologies), 0.5 mM Glutamax (Life Technologies), 1 mM sodium pyruvate (Nacalai tesque), and 1% penicillin/streptomycin (Nacalai tesque) at 37°C with 5% $CO_2$.

### HyFlnD screening

HyFlnD screening followed the protocols described in our previous work with a few modifications (*Takikawa et al., 2014*). A synthetic gene encoding the ε subunit of $FoF_1$-ATP synthase of thermophilic *Bacillus* PS3 was obtained from Genescript, and inserted into the NdeI/XhoI site of a modified pET23a vector (Novagen) with an $8 \times$ His tag. Cysteine mutants of the ATP-binding protein were comprehensively generated by inverse PCR for site-directed mutagenesis. The pair of primers used in the inverse PCR was designed in a back-to-back orientation, and the original codon at the 5′-end of the forward primer was replaced with a cysteine codon. After 5′-kination using T4 polynucleotide kinase (Toyobo), the PCR products were recircularized with a DNA ligation kit ver.2.1 (Toyobo).

For bacterial production of the recombinant proteins, *Escherichia coli* BL21(DE3) cells (Stratagene) were transformed with the circularized PCR products. The transformed cells were grown overnight in a semisolid medium containing 2% tryptone, 0.5% yeast extract, 8.6 mM NaCl, 2.5 mM KCl, 1 mM NaOH, 20 mM $MgSO_4$, 100 µg/ml carbenicillin and 0.8% agarose (SeaPrep) in a 96-deep-well plate at 30°C. Fifty microliters of the melted medium was added to $2 \times$ YT medium (1.6% tryptone, 1% yeast extract, and 86 mM NaCl) containing 100 µg/ml carbenicillin in a 96-deep-well plate. The bacteria were incubated at 37°C until the $OD_{600}$ reached 0.4–0.8. Further incubation at 25°C for 12 hr was performed in the presence of 0.4 mM isopropyl β-D-thiogalactoside (IPTG) (Nacalai tesque). The cells were harvested by centrifugation and lysed in phosphate-buffered saline (PBS; 137 mM NaCl, 2.7 mM KCl, 1.5 mM $KH_2PO_4$, and 8.1 mM $Na_2HPO_4$; pH 7.4) with lysis mix buffer (10 mM $MgCl_2$, 10 µg/ml lysozyme, 0.027% BriJ 58, and 3.3 U/ml DNase I). For purification of the

recombinant proteins, the cleared lysate was incubated for 45 min in a 96-well filter plate containing TALON metal affinity resin (Clontech), followed by washing with PBS containing 10 mM imidazole (Nacalai tesque).

For fluorophore labeling, the recombinant proteins absorbed on the resin were incubated at 25°C for 45 min with Oregon Green (OG) maleimide (Life Technologies), Alexa Fluor 488 (Alexa488) maleimide (Life Technologies), Cy3 maleimide (GE Healthcare), or SiR650-PEG3-maleimide, which were dissolved in PBS at a concentration of 4 µM. SiR650-PEG3-maleimide was prepared by conjugating a maleimide group to SiR-carboxy (*Lukinavičius et al., 2013*) via a PEG3 linker. The resin was washed with PBS containing 10 mM imidazole, and the fluorescent conjugates were eluted with PBS containing 150 mM imidazole and 0.1% bovine serum albumin.

For the performance test of the fluorescent conjugates, the fluorescence intensity in each well of a 96-well plate was automatically scanned with a custom-built fluorescence plate reader system. OG- or Alexa488-labeled conjugates were excited at 460–480 nm and emission was measured at 495–540 nm. Cy3-labeled conjugates were excited at 535–555 nm and emission was measured at 570–625 nm. SiR650-labeled conjugates were excited at 608–648 nm and emission was measured at 672–712 nm. In the first screening, fluorescence changes upon addition of ATP (Sigma) at a final concentration of 2 mM were measured. In the second screening, fluorescence changes upon addition of ATP at a final concentration of 200 nM were measured.

## Production of ATPOS

*Escherichia coli* BL21(DE3) cells were transformed with plasmids encoding the ATP-binding protein with Q105C mutation (ATPBP-Q105C). The transformed cells were incubated in 2 × YT media containing 100 µg/ml ampicillin at 37°C until the $OD_{600}$ reached 0.5–0.8. Further incubation at 25°C for 16 hr was performed in the presence of 0.4 mM IPTG, followed by centrifugation at 4°C for 15 min at 2,070 g. The harvested cells were lysed by means of a freeze-thaw procedure in PBS containing 1 mg/ml lysozyme and then incubated at 4°C for 15 min in PBS containing 0.1% Triton X100, 10 mM $MgCl_2$, and 3 U/ml DNaseI (Takara). After centrifugation of the cell lysate at 4°C for 30 min at 15,000 g, the collected supernatant was purified by using HiTrap TALON crude (BD Biosciences). The purified ATPBP-Q105C proteins (10 µM) were incubated with Cy3 maleimide (50 µM) in PBS at 25°C for 45 min. The labeled proteins were separated from unreacted dyes on a PD-10 desalting column (GE Healthcare). If further purification was necessary, anion-exchange chromatography was carried out on COSMOGEL IEX Type Q (Nacalai tesque) by applying a linear salt gradient. The chromatography was performed with an AKTA purifier (GE Healthcare) according to the instructions of the manufacturer.

To create the ATPOS complex, which enables the anchoring of ATPOS on neuronal surfaces, ATPOS (4 µM) was biotinylated with $NHS-PEG_4$-biotin (Thermo Scientific) at a concentration of 10 µM in PBS at 25°C for 1 hr, followed by separation from unreacted $NHS-PEG_4$-biotin on a NAP-5 desalting column (GE Healthcare). Biotinylated BoNT/C-Hc was prepared as described previously (*Takikawa et al., 2014*). Alexa Fluor 488 streptavidin (degree of labeling 5, Thermo Scientific), the biotinylated BoNT/C-Hc, and the biotinylated ATPOS were mixed in a buffer (10 mM HEPES, 150 mM NaCl, and 2.5 mM KCl; pH 7.4) at final concentrations of 2 µM, 2 µM, and 6 µM, respectively.

## Characterization of ATPOS in vitro

Absorption spectra were obtained in HEPES-buffered saline (HBS; 50 mM HEPES, 125 mM NaCl, 4 mM KCl, 2 mM $CaCl_2$, and 1 mM $MgCl_2$) containing 0.1% bovine serum albumin at pH 7.4 with a spectrometer (V-550, JASCO) at 25–26°C. Fluorescence spectroscopic measurements were performed in HBS containing 0.1% bovine serum albumin at pH 7.4 with a spectrofluorometer (FP-6500, JASCO). The excitation wavelengths of Cy3 and Alexa488 were 550 nm and 490 nm, respectively, and the emission wavelengths were 570 nm and 519 nm, respectively. Fluorescence intensities were measured three times at each concentration of ATP, ADP (Sigma), AMP (Sigma), adenosine (Sigma), CTP (Apollo Scientific), GTP (Nacalai tesque), or UTP (Combi-Blocks) at 25–26°C. Mean values of the triplicate measurements were calculated for each sample. The apparent dissociation constant and Hill coefficient were determined by fitting to the Hill equation. For the assessment of pH dependence, HEPES (pH 7.0–8.5) or MES (pH 6.0–6.5) was used as a buffer compound. The fluorescence

quantum yield (QY) was determined by using Rhodamine B in basic ethanol (QY = 0.65) as a standard (*Kubin and Fletcher, 1982*).

Measurements for the kinetic analysis were performed with a stopped-flow apparatus (SFM 2000, BioLogic), whose flow path was pretreated with a solution (155 mM NaCl, 3 mM $Na_2HPO_4$, and 1 mM $KH_2PO_4$; pH 7.4) containing 1% gelatin (Sigma) or with Bullet Blocking One (Nacalai tesque) for blocking, followed by extensive washing with Milli-Q water. For measuring association kinetics, equal volumes of HBS with 0.1% bovine serum albumin containing either ATPOS or ATP were mixed at ~20 nM final concentration of ATPOS. For measuring dissociation kinetics, HBS containing 5 nM ATPOS and 200 nM ATP was diluted with an equal volume of HBS. Cy3 was excited at 546 nm and the emission was collected through a 570–625 nm band-pass filter (Olympus). Fluorescence intensity was measured five times at each concentration of ATP at 25–26˚C. Averaged traces of fluorescence intensity, *F*, were fitted with the following equation:

$$F = c_0 + c_1 t + c_2 e^{-kt} \quad (c_n : \text{constant})$$

where *k* is the observed rate constant.

## Imaging of cultured neurons

Cultured neurons were incubated with ATPOS complex at a final ATPOS concentration of 600 nM in physiological salt solution (50 mM HEPES, 125 mM NaCl, 2.5 mM KCl, 2 mM $CaCl_2$, 1 mM $MgCl_2$, and 25 mM glucose; pH 7.4) containing 0.1% bovine serum albumin at room temperature. After 10 min incubation, the cells were washed four times with physiological salt solution containing 0.1% bovine serum albumin.

Fluorescence images were acquired with an inverted microscope (IX-71, Olympus) equipped with an EM-CCD camera (iXon EM+, Andor) and a 75 W xenon light source (U-LH75XEAPO, Olympus), using a × 40/0.95 NA dry objective lens (UPlanSApo, Olympus). Cy3 was excited at 520–550 nm and the emission was measured at wavelengths longer than 580 nm with a filter set (Olympus). Alexa488 was excited at 460–480 nm and the emission was measured at 495–540 nm with a filter set (Olympus). The extracellular solution was physiological salt solution containing 0.1% bovine serum albumin and 100 µM ARL 67156 (Sigma) at pH 7.4.

## Imaging in vivo

Male C57BL/6 mice (SLC; postnatal 1–2 months old) were anesthetized with an intraperitoneal injection of a mixture of medetomidine (Nippon Zenyaku; 0.75 mg/kg), midazolam (Sandoz; 4 mg/kg), and butorphanol (Meiji Seika Pharma; 5 mg/kg). The depth of anesthesia was assessed by tail pinch. The body temperature was maintained at 37˚C with a heating pad (BWT-100A, Bio Research Center) throughout the experiment. After the scalp was removed, a custom-made metal frame was attached to the exposed skull with dental acrylic (Fuji LUTE BC, GC) for head fixation. A 4-mm-diameter craniotomy centered 3 mm posterior to the bregma and 3 mm lateral to the midline was performed with a dental drill. The dura was left intact during the procedure. The ATPOS complex solution was pressure-injected into the cerebral cortex for 20–30 min through a glass micropipette whose tip was inserted to a depth of 300 µm from the pia. The surface of the cortex was covered with artificial cerebrospinal fluid (aCSF; 125 mM NaCl, 4.5 mM KCl, 1.25 mM $NaH_2PO_4$, 26 mM $NaHCO_3$, 2 mM $CaCl_2$, 1 mM $MgCl_2$, and 20 mM glucose; pH 7.4).

Fluorescence images were captured with a wide-field microscope (M165FC, Leica) equipped with a 150 W xenon lamp with a high-speed scanning polychromatic light source (C7773, Hamamatsu Photonics) and an EM-CCD camera (Evolve 512, Roper Scientific), using a × 1.0 objective lens (Plan APO, Leica). Cy3 and Alexa488 were excited at 555 nm and 490 nm, respectively, and the emissions of Cy3 and Alexa488 were sequentially collected at 2.5 Hz (200 ms duration for each emission) through a filter set (Leica) at 592–682 nm and at 505–540 nm, respectively.

Two-photon ATP imaging was performed using a two-photon laser scanning microscope (Leica TCS SP8 MPO, Leica) equipped with an optical parametric oscillator (OPO) laser system and a HyD detector (Leica), using a × 25/0.95 NA water-immersion objective lens (HCX IR APO, Leica). Cy3 was excited at 1,045 nm and the emission was collected through a filter set (Semrock) at 565–605 nm.

Test compounds or reagents dissolved in aCSF were pressure-injected at 1 psi into the cerebral cortex with a glass micropipette (10 µm inner tip diameter) inserted at the depth of 300 µm from the

surface. For ATP application, 10 mM ATP was pressure-injected for 10 s. In some experiments, 100 U/ml apyrase (Nacalai tesque) or bovine serum albumin at a concentration equivalent to that of apyrase was pressure-injected for 15 min. For electrical stimulation, a monopolar tungsten microelectrode (1–3 MΩ, FHC) was placed below the dura, and a ground electrode was attached to the custom-made metal frame for head fixation. A train of 100 µs pulses at 200 Hz lasting for 1 s with an intensity of 10 mA was delivered using a stimulus isolator (ISO-Flex, AMPI).

## Data analysis and statistics

Image data were analyzed with ImageJ software (NIH; ver. 1.50e). After background fluorescence was subtracted from images, fluorescence response ($R/R_0$ or $F/F_0$) was calculated relative to the baseline before ATP application or electrical stimulation. All data are presented as mean ± SEM. Wilcoxon's rank-sum tests are used to determine statistical significance.

## Acknowledgements

This work was supported by Grants-in-Aid for Scientific Research (KAKENHI) from the Ministry of Education, Culture, Sports, Science, and Technology of Japan (MEXT) (18K14915 to H Sekiya, 17H04764 and 18H04726 to DA, 19K16251 to H Sakamoto, 18H04609 and 19H05414 to K Hanaoka, 17K08584 to SN, 25221304 to MI, and 17H04029 and 19K22247 to K Hirose), Japan Science and Technology Agency (PRESTO, JPMJPR17P1 to DA), and Takeda Science Foundation (to NK).

## Additional information

### Funding

| Funder | Grant reference number | Author |
| --- | --- | --- |
| Ministry of Education, Culture, Sports, Science, and Technology | 17H04029 | Kenzo Hirose |
| Ministry of Education, Culture, Sports, Science, and Technology | 19K22247 | Kenzo Hirose |
| Ministry of Education, Culture, Sports, Science, and Technology | 25221304 | Masamitsu Iino |
| Ministry of Education, Culture, Sports, Science, and Technology | 18K14915 | Hiroshi Sekiya |
| Ministry of Education, Culture, Sports, Science, and Technology | 17H04764 | Daisuke Asanuma |
| Ministry of Education, Culture, Sports, Science, and Technology | 18H04726 | Daisuke Asanuma |
| Ministry of Education, Culture, Sports, Science, and Technology | 19K16251 | Hirokazu Sakamoto |
| Ministry of Education, Culture, Sports, Science, and Technology | 18H04609 | Kenjiro Hanaoka |
| Ministry of Education, Culture, Sports, Science, and Technology | 19H05414 | Kenjiro Hanaoka |
| Ministry of Education, Culture, Sports, Science, and Technology | 17K08584 | Shigeyuki Namiki |
| Japan Science and Technology Agency | JPMJPR17P1 | Daisuke Asanuma |

Takeda Science Foundation Nami Kitajima

The funders had no role in study design, data collection and interpretation, or the decision to submit the work for publication.

## Author contributions

Nami Kitajima, Conceptualization, Formal analysis, Funding acquisition, Investigation, Visualization, Methodology, Writing - original draft; Kenji Takikawa, Conceptualization, Investigation, Methodology; Hiroshi Sekiya, Conceptualization, Funding acquisition, Investigation, Methodology; Kaname Satoh, Investigation; Daisuke Asanuma, Funding acquisition, Investigation; Hirokazu Sakamoto, Kenjiro Hanaoka, Resources, Funding acquisition; Shodai Takahashi, Yasuteru Urano, Resources; Shigeyuki Namiki, Conceptualization, Resources, Funding acquisition, Methodology; Masamitsu Iino, Conceptualization, Supervision, Funding acquisition, Methodology; Kenzo Hirose, Conceptualization, Supervision, Funding acquisition, Visualization, Methodology, Project administration, Writing - review and editing

## Author ORCIDs

Nami Kitajima (iD) http://orcid.org/0000-0001-9838-832X
Kenjiro Hanaoka (iD) http://orcid.org/0000-0003-0797-4038
Shigeyuki Namiki (iD) https://orcid.org/0000-0003-1520-8261
Masamitsu Iino (iD) http://orcid.org/0000-0001-6426-4206
Kenzo Hirose (iD) https://orcid.org/0000-0002-8944-6513

## Ethics

Animal experimentation: All procedures used in animal experiments were in accordance with the guidelines established by the Animal Welfare Committee of the University of Tokyo (Medicine-P10-010, Medicine-P15-017 and Medicine-P19-092).

## Decision letter and Author response

Decision letter https://doi.org/10.7554/eLife.57544.sa1
Author response https://doi.org/10.7554/eLife.57544.sa2

## Additional files

### Supplementary files

• Transparent reporting form

### Data availability

All data generated or analyzed during this study are included in the manuscript and supporting files.

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
