## [Decision Letter]

**Acceptance summary:**

The manuscript by Kitajima et al. deals with the characterization of ATPOS, a high-affinity optical sensor for extracellular ATP imaging. The centrepiece of this study is an elegant strategy employed to identify sensitive ATP sensors consisting of a small-molecule fluorescent dye conjugated to an ATP-binding protein. This is an important and nicely designed study, which introduces a useful new tool for basic biomedical research.

**Decision letter after peer review:**

Thank you for submitting your article "Real-time in vivo imaging of extracellular ATP in the brain with a hybrid-type fluorescent sensor" for consideration by *eLife*. Your article has been reviewed by three peer reviewers, and the evaluation has been overseen by a Reviewing Editor and Philip Cole as the Senior Editor. The following individual involved in review of your submission has agreed to reveal their identity: Fabio Mammano (Reviewer #3).

The reviewers have discussed the reviews with one another and the Reviewing Editor has drafted this decision to help you prepare a revised submission.

1) ATP is an important extracellular signaling molecule, playing diverse roles in purinergic signaling. In this manuscript, Kitajima et al. make a fluorescent reporter for imaging extracellular ATP, denoted ATPOS. While there are fluorescent reporters for imaging intracellular ATP, it has been challenging to image extracellular ATP with them. This is because the ATP binding affinities are too low to image the extracellular ATP which is in the nanomolar to μM range. Further the pH sensitivity of fluorescent proteins has proved confounding when seeking to image extracellular ATP with existing reporters, because ATP fluctuations in tissues are often accompanied by pH changes.

The present reporter, ATPOS, solves this problem. It is derived from a subunit of the FoF1-ATP synthase of thermophilic Bacillus PS3, which has high affinity for ATP. Using an elegant high-throughput screening strategy, the authors identified a variant where the conjugation of a fluorphore led to fluorescence enhancement in the presence of ATP. In combination with small-molecule fluorophores, it provides a ratiometric readout with a fairly large, pH-independent, fluorescence response. ATPOS can potentially be applied to study purinergic signaling in cancer, immunology, neurotransmission and neuron-glia interactions among others.

2) The major comments from the three reviewers were to clearly discuss the limitations of ATPOS. The questions below provide a framework for this discussion. Clarifications on the Kd, specificity to ATP, time-scales for stable plasma membrane localization, capacity for quantification (a demonstration is not needed). And to consider reframing the claims based on in vivo imaging.

Major questions:

1) The authors put forward convincing arguments for the utility of ATPOS, in particular relating to the advantageous photophysical properties of fluorescent dyes. In addition, we encourage discussing limitations of the new sensor. This would benefit users assessing different complementary tools for their specific applications. For example, genetically expressed sensors require less invasive procedures for sensor delivery in vivo and can be targeted to specific cells and sub-cellular compartments. Can the authors suggest strategies to obtain cellular specificity with their hybrid sensor?

2) Related to kinetic properties, e.g. in Figures 3C and 4C, can the authors comment on how the fluorescence decay of ATPOS follows a drop in extracellular ATP concentration? This information is relevant for the definition of "real-time imaging" and will be important in order to derive physiological interpretations from imaging data.

3) A significant concern regards the quality of the brain imaging results, which should be definitely improved. In particular, the use of single-photon wide-field fluorescence imaging falls short of providing the required cell-level resolution. Even the only two-photon imaging data, presented in Figure 4—figure supplement 1, are rather disappointing. At the current scale and resolution, the claim that "The increase of fluorescence intensity of ATPOS propagating in layer 2/3 was observed with cellular-level resolution" is unsubstantiated. It is strongly recommended to perform better imaging experiments in mouse brains counterstained with a spectrally compatible neuronal fluorescent reporter (e.g. neuron-targeted miRFP703). Otherwise, at the very least, delete the unsubstantiated claim.

---

## [Author Response]

1) The authors put forward convincing arguments for the utility of ATPOS, in particular relating to the advantageous photophysical properties of fluorescent dyes. In addition, we encourage discussing limitations of the new sensor. This would benefit users assessing different complementary tools for their specific applications. For example, genetically expressed sensors require less invasive procedures for sensor delivery in vivo and can be targeted to specific cells and sub-cellular compartments. Can the authors suggest strategies to obtain cellular specificity with their hybrid sensor?

We have added a paragraph to discuss limitations of ATPOS to the main text. We have also shown a possible strategy for cell type-specific targeting of ATPOS in the paragraph (Discussion, third paragraph).

2) Related to kinetic properties, e.g. in Figures 3C and 4C, can the authors comment on how the fluorescence decay of ATPOS follows a drop in extracellular ATP concentration? This information is relevant for the definition of "real-time imaging" and will be important in order to derive physiological interpretations from imaging data.

We have performed an additional experiment in which fluorescence decay of ATPOS upon a decrease in ATP concentration from 200 nM to 100 nM was measured with a stopped-flow apparatus. The decay rate constant was calculated at 2.0 s^-1^, suggesting that the dissociation kinetics between ATPOS and ATP is fast enough to monitor a drop in ATP concentration on a time scale of seconds. We have shown the additional data in Figure 1—figure supplement 1E. We have also added the explanation related to Figure 1—figure supplement 1E to the manuscript (subsection “Generation of high-sensitivity ATP sensors for extracellular ATP imaging”, third paragraph; subsection “Characterization of ATPOS in vitro”, last paragraph; Figure 1—figure supplement 1E legend).

3) A significant concern regards the quality of the brain imaging results, which should be definitely improved. In particular, the use of single-photon wide-field fluorescence imaging falls short of providing the required cell-level resolution. Even the only two-photon imaging data, presented in Figure 4—figure supplement 1, are rather disappointing. At the current scale and resolution, the claim that "The increase of fluorescence intensity of ATPOS propagating in layer 2/3 was observed with cellular-level resolution" is unsubstantiated. It is strongly recommended to perform better imaging experiments in mouse brains counterstained with a spectrally compatible neuronal fluorescent reporter (e.g. neuron-targeted miRFP703). Otherwise, at the very least, delete the unsubstantiated claim.

We have deleted the unsubstantiated claim according to the reviewers’ suggestion.